# Model-Free Control of a Soft Pneumatic Segment

**DOI:** 10.3390/biomimetics9030127

**Published:** 2024-02-21

**Authors:** Jorge Francisco García-Samartín, Raúl Molina-Gómez, Antonio Barrientos

**Affiliations:** Centro de Automática y Robótica (UPM-CSIC), Universidad Politécnica de Madrid—Consejo Superior de Investigaciones Científicas, José Gutiérrez Abascal 2, 28006 Madrid, Spain; jorge.gsamartin@upm.es (J.F.G.-S.); raul.molinag@alumnos.upm.es (R.M.-G.)

**Keywords:** soft robots, soft arm, pneumatic robot, machine learning, neural networks, model-free control, data-driven control

## Abstract

Soft robotics faces challenges in attaining control methods that ensure precision from hard-to-model actuators and sensors. This study focuses on closed-chain control of a segment of PAUL, a modular pneumatic soft arm, using elastomeric-based resistive sensors with negative piezoresistive behaviour irrespective of ambient temperature. PAUL’s performance relies on bladder inflation and deflation times. The control approach employs two neural networks: the first translates position references into valve inflation times, and the second acts as a state observer to estimate bladder inflation times using sensor data. Following training, the system achieves position errors of 4.59 mm, surpassing the results of other soft robots presented in the literature. The study also explores system modularity by assessing performance under external loads from non-actuated segments.

## 1. Introduction

The emerging field of soft robotics offers remarkable promise for revolutionising various sectors, including healthcare [1,2], inspection and exploration [3], and manipulation [4]. Soft robots, inspired by the inherent flexibility and adaptability of natural organisms, exhibit unparalleled advantages in navigating complex and dynamic environments. Unlike their rigid counterparts, soft robotic systems are low-cost, can conform to irregular surfaces, withstand impacts, and can interact safely with humans [5]. However, these advantages come with their own set of challenges, particularly in the realms of sensing and control.

As far as sensors are concerned, the main problem in soft robotics is the need for them to be soft as well. Specifically, in [6], sensors are considered suitable if they offer sufficient compliance without impeding the soft device’s properties, possess resilience and extensibility to withstand multiple motion cycles, and avoid features that could act as stress concentrators and cause damage to the robot.

Although these types of sensors have achieved very good results in terms of accuracy—even being used for applications such as detection of cardiac activity or vocal cord motion [7]—their modelling is highly complex, as they usually present abundant nonlinearities and phenomena such as hysteresis. Additionally, they typically provide indirect measurements rather than directly quantifying the target magnitude [8], like pressure in pneumatic bladders or arm bending, further complicating their analysis.

On the control side, apart from managing such sensors, the sensors must contend with a theoretically infinite number of degrees of freedom and the challenge of characterising the robot’s state [9]. This complexity arises from the abundant nonlinearities between the actuator commands and the robot’s end position. Unlike in rigid robotics, the actuation spaces and configurations do not coincide with each other [10].

Different solutions have been proposed. Model-based methods rely on either simplified models—mainly, the Piecewise Constant Curvature (PCC) hypothesis—which do not achieve adequate precision for many types of robots, or adopt the Finite Element Method (FEM), for which the computational cost makes it very difficult to implement in real time [11,12].

On the other hand, model-free methods allow researchers to achieve adequate precision (errors of about 5 mm are actually achieved) without having to worry about modelling the physical properties of the robot materials, which are in most of cases non-homogeneous and very difficult to characterise with suitable precision. A wide range of Machine Learning (ML) techniques have been employed: most notably, the use of different types of neural networks [13] and reinforcement learning [14]. The big bottleneck of these methods is often the need for a sufficiently large dataset.

In this work, closed-loop control that gives position errors lower than 5 mm has been implemented over a soft pneumatic segment that is 150 mm long. As far as has been investigated, and apart from the difficulties that are always involved when comparing manipulators with such different actuators, these are the best results for measuring the error/length ratio of a manipulator that have been reported in the literature for model-free controllers.

To achieve this level of performance, there are two main contributions of this paper:The characterisation and implementation of three resistive sensors in a segment of the robot;The development of a closed-loop control loop based on feedforward neural networks that achieves very low error.

More specifically, it can be said first that resistive sensors have been inserted into the soft robotic arm presented in [15]. These are made of carbon-black impregnated rubber and have piezoresistive effects: they react by changing their resistance based on variations in length. They have been characterised and found to behave independently of temperature. However, due to the inability to establish a precise theoretical model accounting for hysteresis and saturation, these sensors required modelling through a neural network.

Simultaneously, a closed-loop position control system for one of the robot’s segments was developed; it operates based on bladder inflation times. In this scheme, the desired position is translated into inflation times within the feedforward module. This module consists of an experimentally trained neural network that correlates inflation times with corresponding positions. Subsequently, the actual inflation times, estimated by the previously characterised sensor network, are compared with these reference times.

Unlike other robots in the literature, the use of a Feedfowrard Neural Network (FFNN) architecture, which, in contrast to Long Short Term Memory (LSTM) networks or other architectures, does not need to take into account previous robot positions during training, gives sufficiently accurate results. This reduces the number of samples needed to achieve correct training, which, when working with a physical robot and not a simulator, is a considerable advantage.

The system’s robustness was evaluated by introducing additional identical segments to the robot, enabling an assessment of both its modularity and its performance under added weight, which remains a prominent challenge in the realm of soft robotics. While potential enhancements exist, the results obtained demonstrate satisfactory performance under these conditions.

The paper is structured as follows: Section 2 presents the most recent works on soft sensors and control of soft robots. Section 3 describes the different steps done until the control loop was designed, and Section 4 details the tests carried out and the results they produced. The results are commented on and compared to those achieved in other works from the literature, and the validity of the model when external loads are applied is discussed. Finally, Section 5 draws the conclusions.

## 2. Related Works

### 2.1. Sensing

While in cable-actuated soft robots it is possible to use the motor’s sensors—such as its encoders, which are in charge of tensioning the cables—to close the control loop [16], this is not the case in pneumatic, hydraulic or SMA-based robots, where it is usual to resort to indirect sensing [8] by using non-metallic materials with some kind of property of interest that is correlated to the action taken on the system.

#### 2.1.1. Capacitive Sensors

Different natures of sensors have been employed. Because they are common in traditional robotics but also very adaptable to these new manipulators, capacitive sensors have been used in soft robots. They are used for human motion detection [17], wearable electronics [18] and haptic applications [19]. As stated in [20], their main advantage is that the capacitance only depends on geometry and not on the conductivity of the elastomer, which makes theoretical modelling more accurate when describing their behaviour. Nevertheless, their capacitance is extremely dependent on temperature and humidity, which, unless they are to be used to measure these quantities directly, is a major disadvantage [21].

Capacitive sensors have been used as pressure sensors by employing MXenes: a recently discovered category of two-dimensional materials that combines a transition element—such as Ti, V, Nb or Ta; denoted by M—and a transition element—C or N; represented by X [22]. Different layers of the material are stacked and are adhered using silicone or hydrogel, thereby causing the material behave like several capacitors in series. In [7], an elastomeric salt matrix is used in conjunction with ${\mathrm{Ti}}_{3}C_{3}2T_{x}$ MXene foils—thus achieving high accuracy in pressure measurements, even for the most discrete movements such as vocal cords when speaking or the heartbeat.

Another possibility to manufacture capacitive sensors is to use, as done in [23], elastic fibres filled with liquid metal. By deforming the sensor, a change in the capacitance is achieved by the movement of the liquid metal inside the fibres.

#### 2.1.2. Resistive Sensors

Resistive sensors are also commonly found and are based on the change to their resistivity as their elongation or the pressure or temperature exerted on them varies. Their usage has been proven to be adequate in soft hands [20], grippers [24], human motion tracking [25,26,27] and soft mobile robots [28] due to their excellent mechanical and electrical properties [29] and their high resolution [30].

Physically, they consist of the insertion of conductive microparticles—usually carbon—on a polymeric matrix [31], which allows a high working range—there are sensors that can operate under compression without any problem [32]—and, in some cases, anti-freezing or self-repairing properties [33].

Although their manufacturing process tends to be simpler than that of capacitives, no standard modelling methodologies exist due to the wide discrepancies in their behaviour. Indeed, sensors can range from highly linear [34] to some with considerable dynamic effects [35].

Abundant models derived based on the microscopic properties of materials [17,19,36,37] are available. The major challenges they have to face are negative piezoresistivity and hysteresis.

The first phenomena, despite being counter-intuitive, is widely found in numerous sensors [38,39,40] and has been discussed in theoretical works since the end of the last century [41]. Nevertheless, modelling the resistivity coefficient in those cases continues to be an open challenge. Along the same line, not many articles are able to provide an analytical explanation of hysteresis with adequate precision [42], and it is preferred to resort to polynomial adjustments [43] or simply to know what the maximum error will be as a consequence of hysteresis and to see if it is within an acceptable range [44]. As commonly occurs in other areas, ML techniques are also employed [45].

#### 2.1.3. Other Sensor Typologies

Because they have been widely used in rigid robotics and offer a highly dynamic range and the possibility of obtaining linear relationships between the entry and output, inductive sensors are often considered despite their size and stiffness, which are usually too high for implementation in soft robots. One example can be found in [46], where three spring coils are located inside a pneumatic silicone module and are used to measure its bending based on the change to the inductance that the coil undergoes when deformed.

Eddy-current-based sensors, on the other hand, can be used for pressure and force measurements due to their low hysteresis, good repeatability, high robustness to atmospheric pollutants and high sensitivity [47]. In [48], a sensor array achieves force measurements with resolutions of less than 1 mN over a range of 15 N by measuring the eddy currents of each sensor.

Finally, among the more innovative options, it is worth highlighting the existence of Hall effect sensors capable of measuring force embedded in silicon matrices [49] and optical sensors that can measure deformations in a soft robotic hand [50].

### 2.2. Control

The theoretically infinite degrees of freedom of robots make their control complex and also have to take into account possible redundancies.

#### 2.2.1. Model-Based Control

A first option is the use of model-based control techniques. These allow better understanding of the system, greater predictability and adaptability, better management of uncertainties, and the possibility of working in contact situations between the robot and its environment [11]. In addition, they achieve efficient workspace determination, where the error in their definition can be determined a priori [51].

A first approach in this line is the usage of the PCC hypothesis, which relies on each section of the robot deforming along an arc of circumference and assumes the absence of gravity. Even though these are very demanding conditions that are usually suitable for hyper-redundant or rigid continuous robots [52] but not for soft robots, there have been robots that have performed with enough precision following them.

In [53], a continuous and flexible robot is matched with its augmented rigid body model, which can be formulated with PCC. From there, a state space model is generated from which to generate feedback that achieves good closed-chain control for the first two segments of the robot. In [54], the same formulation is used to develop a Model Predictive Controller (MPC), which is robust against perturbations although it cannot avoid high tracking errors (5 cm). Finally, ref. [55] uses PCC-based control for a robot that interacts with the environment.

FEM-based models, on the other hand, do not need to rely on such stringent assumptions as those of PCC. They are a widely used tool in other areas of engineering and have proven to also be suitable for modelling nonlinear soft materials and their interactions with the environment. Although there are many successful examples of soft robot modelling and simulation using this technique [56,57,58], with errors near 5 mm, its large computational cost often makes its use in real-time closed control loops unfeasible. It is necessary, if acceptable simulation times are to be achieved, to use reduced-order models, such as done in [59,60,61].

If low times are not a major requirement, remarkable results can be achieved using FEM. Thus, in [62], an open-loop controller with the ability to regulate robot stiffness is proposed, and [63] proposes an FEM-based nonlinear controller for a silicone robot actuated by cables. In the model, hysteresis and friction of the manipulator are considered—although not much precision is needed modelling them—and good perturbation rejection is achieved. Before [64], the same author had explored the usage of a gain scheduler combined with an FEM simulation for the same robot.

#### 2.2.2. Model-Free Control

In contrast to model-driven methods, there is also a very broad research track that uses, or relies on, ML techniques for the control of these robots. These do not need to impose restrictive assumptions, allow real-time results and are capable of capturing phenomena associated with the robot’s manufacture or the materials used, which are normally very difficult to model in a sufficiently accurate way [65]. ML techniques have long been used for the modelling of a wide range of traditional robot typologies, including manipulators and parallel and mobile robots [66,67,68,69].

A first approach is the usage of sim-to-real models. These are models trained from simulation data, from which a sufficiently large number of samples can be realised without leakage or puncture problems, which are then adjusted in a second stage to the real robot. This is the approach followed by [16], which trains an FFNN from 28,800 samples extracted from the PCC model of its soft robotic neck with errors of 10% of the robot workspace.

While these models achieve correct results, it may seem of more interest to train from FEM models in order to combine their accuracy with the fast response that a finite element simulation is not able to provide. Thus, in [70], PCC parameters are extracted from an FEM model and used for training an FFNN and making open-loop control of the robot. In [71], two FFNNs are placed in series: a first one trained using an FEM model (which requires 4000 points for correctly modelling a soft segment) and a second one trained over the real robot, for which 300 samples are enough; the authors obtained tracking errors of 1 mm. This approach allows researchers to work, in addition, with the Jacobian matrix of the manipulator without the need to estimate it by deriving the net.

Other authors directly train the neural network from real data, as is the case in [72], where 3430 points are required (a reasonable number) for achieving accuracies of 4.42 mm. In [73], the Jacobian matrix is learned from real data. Nevertheless, because bad predictions are achieved near singularities, the authors propose to include in the control loop the direct kinematics model, which serves to correct the learned inverse kinematics for those points, reducing error from 21 to 4 mm. Not only is FFNN used in learned models, the use of the Koopman operator is starting to become widespread in soft robot modellings [74] and has already been applied, albeit with moderate success, on some manipulators [75].

The problem with data-based approaches is that they do not naturally allow the management of redundancies; however, they often appear in soft robots as a consequence of their high dimensionality. One solution, proposed in [76], is to work not only with the final tip position but with the whole manipulator. A Convolutional Neural Network (CNN) is employed to read the robot’s current position and generate the pressures necessaries to reach the desired position. The control loop, however, remains slow: taking 30 s to reach each new target. In a similar way, [77] combines visual and tactile sensing to estimate the robot’s position.

Finally, Reinforcement Learning (RL) is also being used to control soft manipulators. In [78], the agent—the soft manipulator—is modelled using a Nonlinear Autoregressive Network with Exogenous Inputs (NARX), and trajectory optimisation is included. The work achieves stable open-loop control of a very chaotic system (but with errors of the order of cm), which is closed in [79] by using an MPC to achieve 9 mm of tracking error. Ref. [80] uses RL to make a soft robot able to work under different payload conditions. Going one step further, the authors of [81] proved the robustness of RL controllers when changing the geometry, velocity or materials of the robot.

### 2.3. PAUL Approach

As presented in the Introduction, the robot on which the closed-loop control has been implemented has already been presented in a previous work [15]. As can be seen in Figure 1, it is a modular, pneumatically actuated robot made up of different silicone segments. Manufacturing is done by casting silicone, although other technologies such as 3D-printed silicone are being considered for future lines in order to increase repeatability.

Each segment is 100 mm long, weighs 187 g and has three bladders, giving it three degrees of freedom. The PAUL inputs are the inflation times of each bladder. The position is measured using a beacon consisting of three coloured balls and two cameras.

The primary aim of this work is to attain closed-loop kinematic control of one of the robot’s segments with a point-to-point error of less than 5 mm. If this requirement is met, the subsequent objective will be to evaluate this control objective when the weight of additional segments is carried by the robot.

In the design stage, grooves were provided on PAUL’s surface in which some kind of sensor could be inserted; we retained the option to place the sensor at the junction between segments if this was finally considered to be more appropriate.

Of the different options studied, the use of capacitive or resistive sensors was considered, inductive sensors being discarded due to their lack of natural rigidity and other options due to their lack of maturity. Although capacitive sensors are, as has been said, easier to model, the most accurate results obtained—such as those presented in [20]—are still insufficient approximations in many cases, which inevitably makes it necessary to resort to the use of ML. In addition, their greater variability in the face of changes to the temperature and/or humidity make their behaviour more unstable, which finally led to the use of resistive sensors.

In terms of control, the use of ML techniques was chosen due to the greater ease of achieving accurate results. As demonstrated during the literature review, model-based methods do not offer higher accuracies and need careful parameter selection. In addition, they are slow, which is detrimental in closed-loop control. Indeed, in addition to modelling the robot’s inflation, it is also necessary to characterise the behaviour of the sensors. As these sensors have negative piezoresistance, their analytical treatment was very difficult.

FFNNs were used for modelling both the sensor and the segment and were trained from real data. More-complex architectures were not used, as they require more training samples to achieve good results. Compared to the use of sim-to-real techniques, only 1000 samples were enough to train the model, which can be done in a reasonable time and shows the capacity and robustness of the control methodology used. The lack of redundancies in a single segment made it unnecessary to combine different types of sensors or to use cameras within the control loop.

## 3. Materials and Methods

### 3.1. Sensor Characterisation

The Adafruit conductive rubber cord stretch, displayed in Figure 2, was used as a sensor. It is a resistive sensor with 2 mm of diameter that, according to the datasheet specifications, is linear and has a resistivity of 150 Ω/cm and an elongation at break limit of 150%.

In order to test the validity of these specifications and to characterise the sensor to work with it, the test rig shown in Figure 3 was designed. It consists of a stepper motor that rotates a rack and pinion mechanism. Attached to the rack was one end of the sensor, which was also connected to ground. At its other end, the sensor was wired to a 24 V power source and, in series, a 10 kΩ resistor.

After each step of the motor, the voltage at the sensor was measured using an Instrumentation Amplifier (INA), model 3221, connected via series to an Arduino Uno which, in turn, dumped the data to Matlab once the process was finished. As the primitive diameter of the pinion and its number of teeth are known, it is also possible to know, at that moment, the length that the rubber has been stretched to utterly establish relationships.

In each measurement cycle, the sensor started contracted, stretched the entire length of the bench rack (12 cm), and after a pause of one second, was made to return to its initial state; the voltage was also measured during this contraction phase. In the first part of the video in Appendix A, one of the characterisation and measurement processes can be seen.

This method was used to characterise both newly arrived sensors and those inserted into the slots PAUL has for them. It was found that the rubber changed its behaviour when it came into contact with the robot’s silicone. Experimentally, it was found that this change in properties was sufficiently noticeable and stable after seven days of permanent contact between the sensor and the segment. It is believed that a chemical reaction occurs between the silicon and the carbon in the sensor that causes this change in behaviour. It has not been explored whether this alteration in properties occurs only with the silicone from which the segment is constructed (TinSil 80-15) or whether it occurs with other silicones.

The results of the characterisation can be seen in Figure 4. The characterisation was carried out with new sensors, i.e., they were only subjected to a reduced number cycles on the test bench. It has been observed that the effects of drift or alteration of properties over time are negligible. In fact, it was found that after 1500 cycles of stretching and relaxation on the robot itself, when the sensors were placed back on the test bench, the graph of results obtained was very similar.

As can be seen, the sensor, once modified by the silicone, performs better. Firstly, its response is much more linear and takes much longer to saturate, which extends the sensor’s measurement range. Secondly, the elongation and compression curves are much more similar and present monoticity, which ensures that there are no two elongation values with the same associated voltage value, unlike what happens in the relaxation curve of the original sensor. Finally, the hysteresis cycle of the latter is much smaller and, in addition, returns when no voltage is applied to the initial voltage. In the original sensor, this does not happen, but it is necessary to wait for a period of time—several minutes according to the datasheet—until it recovers its original state, which would make its use in control loops unfeasible.

The main drawback of this second behaviour is that the piezoresistivity is negative, i.e., the resistance decreases as the sensor is stretched, which makes any form of analytical modelling impossible. An attempt was made to fit, polynomially or exponentially, the outgoing and return curves of the sensors in order to be able to use them for control.

The aim was to be able to correlate the measured voltages with the length and from there to be able to estimate the swelling time of the bladder associated with that sensor. However, it was found during this phase that the swelling of any of the bladders had repercussions on the measurements of the segments associated with the other two, as the deformations it introduced had an effect on the entire segment, not just locally. The sensors were therefore modelled with a neural network. As this process was done in conjunction with the control loop design, it will be explained in Section 3.3.

Furthermore, the modified sensors were compared, and their behaviour was analysed. Figure 5 illustrates that the shape of the curve is consistent across all sensors; however, there are significant variations in the values recorded. This can be attributed to two factors. Firstly, the sensor pieces do not have identical lengths, making it difficult to ensure that they are attached at the same point on the test bench. Additionally, the manufacturer acknowledges that the sensors have high tolerances due to the manufacturing process.

While we initially considered calibrating or scaling the sensors to overlap the hysteresis curves during operation, this was ultimately discarded. As Machine Learning techniques will be used, modelling will be experimental and will be able to capture the unique characteristics of each sensor.

Finally, the independence of the sensor’s behaviour from the temperature was also proven during this phase. An experiment was carried out several times with an infrared heat lamp pointing at the sensor. The temperature was measured at the beginning of each measuring cycle. Temperatures between 23 and 45 °C were studied. This was considered an acceptable range of working temperatures for the robot.

Figure 6a depicts the behaviour of a sensor at different temperatures. It can be observed how all the stretching curves virtually coincide, and, even though it is not the case for the relaxation curves, they are not clearly ordered, which would have suggested strong dependence between the ambient temperature and the behaviour of the sensor.

In Figure 6b, it can be seen how the average value of the voltage measured over the cycle is very similar when the ambient temperature is higher than 29 °C and when it is lower. The red lines, which indicate the median, practically coincide: for the first case, the median voltage is 7.90 V, and for the second one, it is 7.88 V.

### 3.2. Changes to Segment Design

To implement the control system, the first step was to attach the sensors to the segment. As the segment receives three actuation inputs, three sensors were incorporated.

The sensors were inserted into the pre-stressed segment to ensure no dead zone where the tension does not vary at the beginning of the stretch. Figure 7 compares the tension measured by a sensor as its length varies on the test bench with the tension measured by the same sensor implemented on a segment against the estimated stretched length. The graph of the latter is shifted to the left, indicating a decrease in strain from the beginning and earlier saturation. This is not problematic, as the swellings to which the bladders are subjected do not cause significant deformations in the sensor. The curves are not entirely parallel due to the U-arrangement, which causes non-uniform stretching of the sensor.

Although the segments were designed with grooves on their surfaces to accommodate the sensors, some modifications were required during the implantation process.

Firstly, it was found that the sensor did not stay with the segment when it was deformed, resulting in incorrect measurements. To remedy this, it was decided to bond them directly to PAUL using DOWSIL™ 732 multi-purpose sealant. This component is transparent and does not alter the electrical properties of the sensor; however, it requires a curing time of 24 h.

Furthermore, it was found that the sealant could also be used for repairing small punctures and leaks as well as for sealing the tubes and the bottom of the segment, which was done in the previous work [15] by curing silicone. This meant less waste of material, as with this new method, only the three holes in the bladders were sealed rather than a layer of silicone being added.

In addition, the wiring and instrumentation of the sensors had to be managed. The electronic instrumentation was attached to the cables with alligator clips, and the joint was coated to prevent accidental contact with the glue during the sealing of the tubes. Figure 8 shows the segment with the new design features implemented and the wiring details.

The location of the segment’s INAs was also a point of discussion. Initially, the idea was to place them at the junction between segments and from there to use the central hole of the robot to bring only the two wires needed for I2C communication to the Arduino. This option required less wiring and was more robust against noise. The difficulty, however, of introducing the INA there and the decreased accessibility of the INA for solving problems meant that this option was discarded once it was seen that there were no noise problems when transmitting the analogue signal from the sensors inside the robot. Multi-stranded cables from an HDMI cable were chosen as they could be carried inside the robot without impeding its flexing.

### 3.3. Control Loop Design

The control diagram used is depicted in Figure 9. It consists of two feedforward neural networks and a P controller that interact with the segment and the sensor readings. The mission of the first neural network is to convert the target position $x_{ref}$ into bladder inflation times $t_{ref}$, while that of the second is to predict the actual inflation times $t_{est}$ from the sensor voltage readings $u_{mes}$.

Initially, we considered controlling the system by feeding back the robot’s position, i.e., using a network to play the role of state estimator and a second network, instead of the controller, to predict from the predicted position errors. However, the high nonlinearity of the system—which implied very different swelling times to correct an identical error at distant points in the workspace—meant that this possibility was ruled out.

Initially, simpler techniques such as polynomial regression or experimental fits of state functions were considered to for the design of the state estimator. However, the nonlinearity of the estimator and its hysteresis, which, although moderate, is significant, made this option unfeasible. Similarly, it was found that the use of Recurrent Neural Networks (RNNs) did not significantly improve the results while requiring many more samples to achieve correct training of the network as it was necessary to store and input values of past states of the segment.

### 3.4. Data Acquisition and Networks Training

The data necessary for the training of both networks were captured using the PAUL vision system, which has already been presented in [15]. An image of the training process can be seen in Figure 10 and in the second part of the video in Appendix A.

At each point, the data collection process was as follows:A random combination of inflation times was generated. Times were chosen in the interval [0,1000] ms for safety reasons and in steps of 50 ms. Although that discretisation was not strictly necessary, it will be very helpful in the next step.It was checked that the previous combination had not been previously generated. The aim of that verification was to ensure that the training samples were evenly distributed throughout the segment’s workspace. If this was not the case, the system returned to step 1.PAUL was inflated based on the generated value.The position, voltage and inflation time were recorded.The segment was completely deflated to avoid hysteresis effects during the data capture process. When an LSTM net was tested, this step was not done, as previous positions were also used for training.

A total of 700 samples were captured using this methodology. Because pauses were left between each process step to ensure correct inflation and error-free data collection, each point took about 5 s to capture. The total process, therefore, took approximately one hour. This is a very reasonable time that is perfectly manageable and for which the risks of leaks or punctures are small.

The Reference Generator comprises an input layer of size 3 corresponded to the recorded position in *x*, *y* and *z*; a hidden layer of 25 neurons utilising a sigmoid activation function; and an output layer of size 3 with a linear function. The State Estimator mirrors this structure. Nevertheless, as all inputs and outputs should be positive, a Rectified Linear Unit (ReLU) was employed as an activation function for the hidden layer.

In the training process, 80% of the available data were allocated for training purposes, while 10% each were designated for validation and testing. The maximum number of epochs was predefined as 1000; however, training concluded before reaching this limit because 6 validations checks with consecutive increments of the error were achieved.

Figure 11 depicts training curves and error histograms for both networks. As can be observed, the Reference Generator was easier to train than the State Estimator: fewer epochs were required and lower errors were reached. This is probably due to the fact that positions have much more predictable behaviour than voltages, which are affected by sensor hysteresis.

Nevertheless, the results can be considered very acceptable, as the error histograms for both cases show deviations between the real and predicted inflation times on the order of tens of milliseconds. These errors are practically negligible on PAUL, and moreover, it should not be forgotten that the network has been trained with a 50 ms discretisation. Finally, the closeness of the three curves (train, validation and test) indicates that these results imply little risk of network overfitting.

## 4. Results

Three experiments were carried to validate the precision and validity of the control system implemented here. In a first step, PAUL was asked to reach randomly generated points inside its workspace. The distance between the point achieved and the desired point was measured. The second experiment consisted of drawing a triangle and a square. Finally, additional segments were added to PAUL in order to study the weight-carrying capacity and the modularity of the system.

### 4.1. Point-to-Point Movement

In this first experiment, the robot was asked to reach a series of 40 random points in its workspace. For the first half of the points, the robot was completely deflated from point to point, while for the remaining ones, the robot travelled sequentially from one to another. The idea behind this was to measure the accuracy of the controller and to assess whether the training method was correct, or whether, on the contrary, the hysteresis of the manipulator made it necessary to resort to more complex techniques, such as the aforementioned RNNs.

An error in norm 2 and of 4.27 mm with a standard deviation of 2.67 mm was obtained. For the data with intermediate deflation, the errors were reduced to 3.94 mm with a standard deviation of 1.25 mm, while for the second set of test data, the error obtained was 4.59 mm with a standard deviation of 2.35 mm.

Although, as expected, the values of the second set of experiments are somewhat worse, they do not differ much from those performed under the same training conditions. In addition, Figure 12 illustrates how the error does not increase systematically over time, confirming the validity of an FFNN and the lack of need to memorise previous manipulator positions to achieve accurate control. On its side, Figure 13 shows how the control system is valid for the whole workspace—which can be assimilated here to a 10 cm square—since the errors are similar in all of its regions.

Figure 14 presents the iterations performed by the robot during its movement between two points and the controller inputs, for which the maximum actuation value was limited to 300 ms in a single iteration for these iterations. Specifically, the robot moved from the origin to point (16,30,97)
mm, which is reached with swell values of (575,800,0)
ms. It can be observed how bladder 3 practically does not inflate, as expected, and, as for the other two, during the initial iterations, the control signal is high but quickly decreases as it is seeking only to make fine adjustments. The virtually zero overshoot and low settling time make the use of integral or derivative actions unnecessary.

Comparing these results to other results achieved by similar soft manipulators in the literature, it can be said that the proposed controller reaches very low errors. Looking specifically at the error relative to the manipulator length, a PAUL segment with 4.24 mm precision has a relative error of 2.49%. Only FEM-based controllers are able to achieve higher accuracies: close to 1%. In the field of Machine Learning, although the work of [72] achieves errors of 0.9%, this is a cable robot, and therefore, greater precision is expected. Among pneumatic manipulators, only [80], controlled with Reinforcement Learning, achieves similar results.

### 4.2. Figure Drawing

The second experiment consisted of drawing two simple figures with the end of the segment: a triangle and a square. Figure 15 depicts the results obtained. Average errors over the entire path were, respectively, 3.98 and 4.07 mm, and their corresponding standard deviations were 1.76 and 2.02 mm; these results are very similar—as expected—to the errors of the point-to-point experiment without intermediate deflation.

As can be seen, the trajectories taken by the robot are close to the desired trajectories. The tracking of the trajectories is precise except for one of the sides of the square. The biggest error is introduced by small oscillations in the vertices as a consequence of the abrupt change of direction.

This experiment was repeated a second time in order to validate the reproducibility of the control system when drawing figures. The results were very similar: errors of 4.01 mm for the triangle and 3.99 mm for the square were achieved. The respective standard deviations were 1.21 and 1.69 mm.

### 4.3. Multiple Segment Tests

This last experiment aimed to investigate the performance of a controller trained on a single segment when additional modules were integrated into the robot. This test possesses a dual interpretation. Firstly, it serves to assess the payload capacity of PAUL, which has exhibited excellent performance under loads of up to 100 g [15] despite each individual segment weighing 161 g.

Secondly, it facilitates an analysis of PAUL’s modularity: by comparing the positions attained by the end of the initial segment when unladen versus when additional segments are added, insight can be gained regarding the system’s modularity. Consistency in the reached positions would suggest a modular system wherein the influence of new segments on existing ones is minimal. This implies the potential for individual training and control of each segment, which facilitates the construction of the overall system as an amalgamation of distinct subsystems.

The tests consisted of attaching first one and then two additional non-actuated segments to PAUL and asking the controller to position the end of the first module at a position in its workspace. Essentially, the process detailed in Section 4.1 was reiterated with increased weight and length. With one segment, PAUL is 125 mm long and weighs 187 g. When one segment is added, it becomes 250 mm and 375 g, and it is 375 mm and 560 g when two segments are added. The intended position was captured using the green marker illustrated in Figure 16, and the complete trihedron was not required since the orientation data were not recorded.

Twenty points were taken for each situation. In the first one, a mean error of 9.27 mm with a standard deviation of 6.70 mm was obtained, and in the second one, the mean error was 11.83 mm with a standard deviation of 7.03 mm.

These results notably deviate from the accuracies achieved by the controller with a single segment, indicating a significant challenge to realising a modular control approach at initial assessment.

Several factors contribute to this outcome. Firstly, the added weight, already demonstrated to be excessive for open-loop control, is now shown to challenge the capabilities of the closed-chain controller. Additionally, the increased hysteresis of the sensors due to heightened stretching of the segment due to additional weight exacerbates control challenges. Moreover, oscillations observed with two or more modules, which were absent when using a single segment, pose difficulties that the controller struggles to manage.

Despite the complexities associated with these challenges, except for the last one, potential strategies in the design of the control loop could be investigated to mitigate them and enable modular operation with PAUL. One approach may entail the consecutive implementation of two neural networks within both the Reference Generator and the State Estimator: the first one extensively trained on data acquired without load, followed by a second network trained on the specific number of segments in use. The second network is intended solely to fine-tune the outcomes of the preceding network and requires only a few samples. Moreover, the utilisation of Transfer Learning techniques could offer promising results.

## 5. Conclusions

Soft robotics has emerged as a thriving field and boasts numerous advantages and a diverse range of applications. The surge in related research activities in recent years underscores its increasing significance. Nonetheless, several challenges persist, warranting further exploration.

One such challenge pertains to sensor integration. Incorporating sensors into these devices without compromising their soft nature necessitates the exploration of alternatives that diverge from conventional robotics practices. Unfortunately, there are no universally applicable solutions to this predicament. Furthermore, the implementation of high-precision controllers remains a complex task.

This study addresses both of these challenges by integrating sensors onto a pneumatic manipulator segment and establishing a closed-loop control system.

Low-cost resistive sensors were experimented with, and their behaviour was observed, which indicated negative piezoresistivity and substantial hysteresis. Analytical modelling proved challenging due to these characteristics. Additionally, attempts to refine FEM models and the limited accuracy of methods like PCC led to the employment of a neural network to model the relationship between bladder inflation times and the end positions of PAUL.

Consequently, a control loop with a PID regulator was devised, employing the inflation time as the control variable. The Reference Generator FFNN translated the desired position into a reference time, while the State Estimator estimated bladder inflation times based on sensor data.

Training the system required a relatively modest number of samples (700) and was achievable in roughly an hour. Upon completion, we achieved errors of 4.24 mm, representing 2.49% of the robot’s length. The objective of control can be considered fulfilled. Furthermore, the accuracy of the proposed controller is demonstrated by comparing the results with those achieved by other soft robots.

The system’s performance was assessed by adding additional segments. It was found that the developed controller lacks modularity and is unable to fulfil the additional requirements. However, there is potential to extend the training of an unloaded segment to the entire robot, which could offer PAUL significant versatility with minimal training effort. This is a direction for future research aimed at replicating the precision achieved with one segment for multiple segments.

## Figures and Tables

**Figure 1 biomimetics-09-00127-f001:**
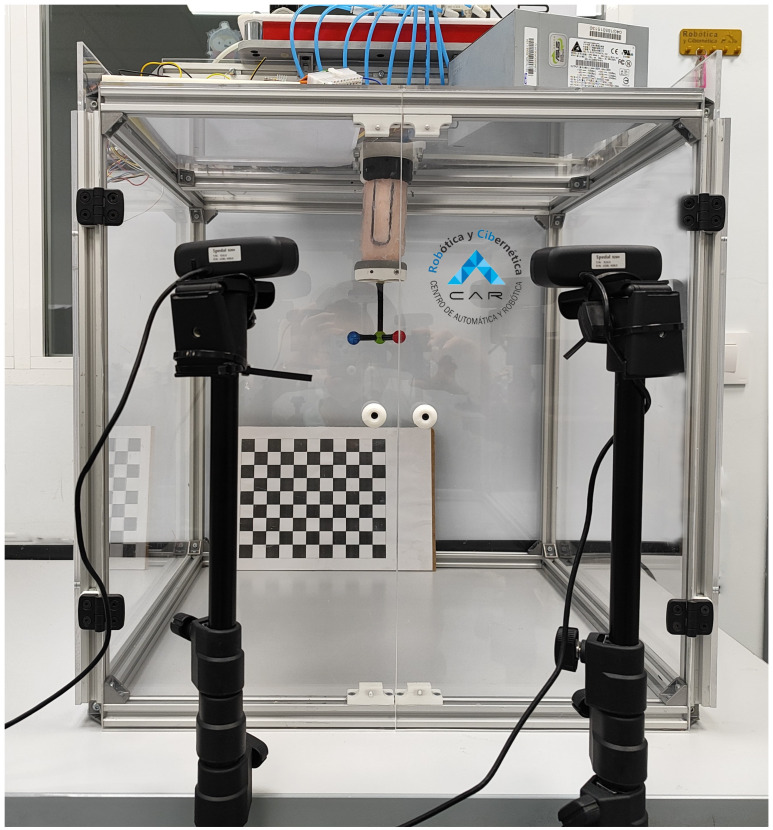
PAUL robot with one segment equipped with the sensors and its working environment. Source: Authors.

**Figure 2 biomimetics-09-00127-f002:**
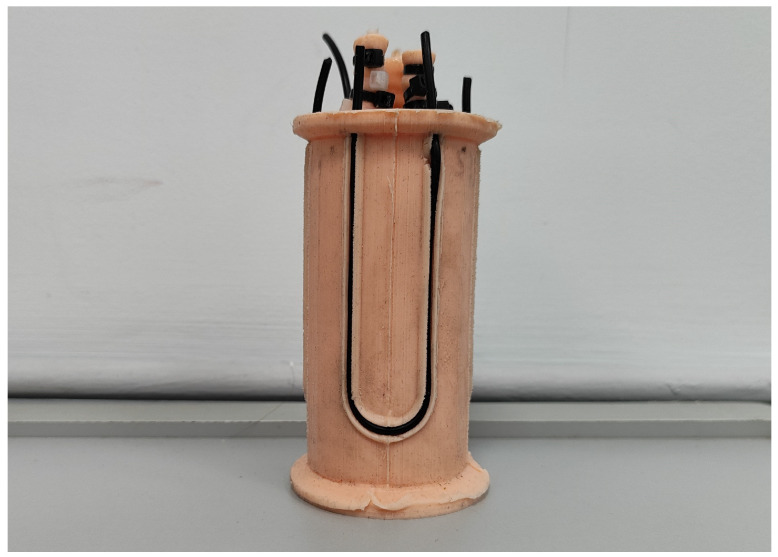
Conductive rubber strain sensor implemented in one of the PAUL segments. Source: Authors.

**Figure 3 biomimetics-09-00127-f003:**
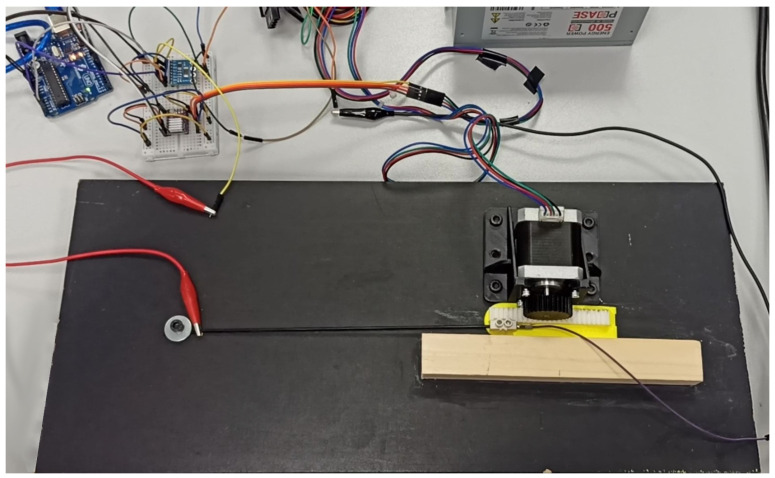
Test rig employed for sensor characterisation. Source: Authors.

**Figure 4 biomimetics-09-00127-f004:**
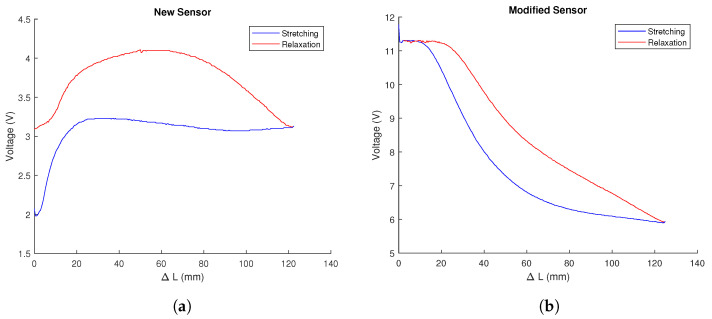
Behaviour of the sensor as a result of its characterisation on the test bench designed for this purpose. The graphs show variation in voltage (in Volts) with respect to an increase in length, expressed in millimeters: (**a**) For a newly arrived sensor. (**b**) For a modified sensor, which presents less hysteresis and higher measuring range. Source: Authors.

**Figure 5 biomimetics-09-00127-f005:**
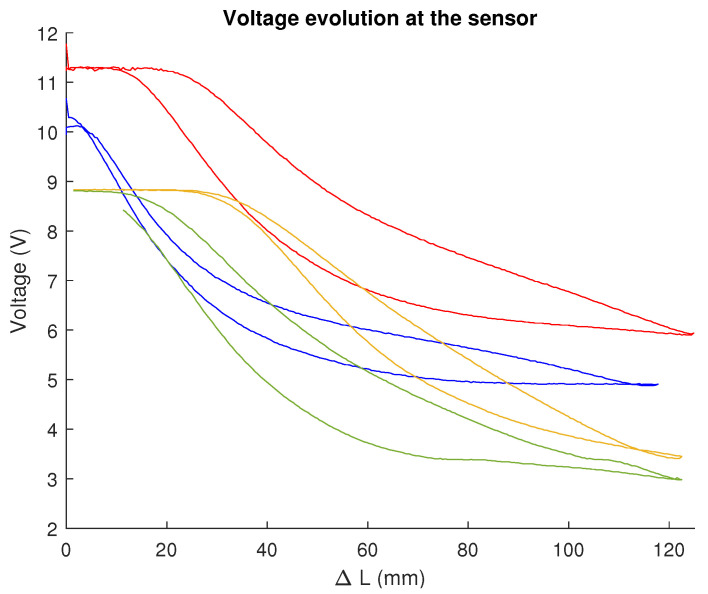
Behaviour at constant conditions for different sensors. Source: Authors.

**Figure 6 biomimetics-09-00127-f006:**
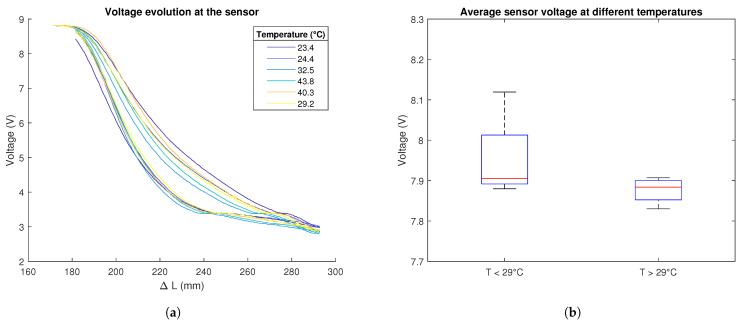
Relation between sensor behaviour and temperature. (**a**) Behaviour of the sensor when the experiment carried out in the test rig is done at different temperatures. (**b**) Box plot diagram of the average sensor voltage over a complete stretching and relaxation cycle for temperatures under and over 29 °C. The red lines indicate the median average voltage for each group, the blue boxes enclose all the data between the Q1 and Q3 values, and the whisker marks minimum and maximum values. Source: Authors.

**Figure 7 biomimetics-09-00127-f007:**
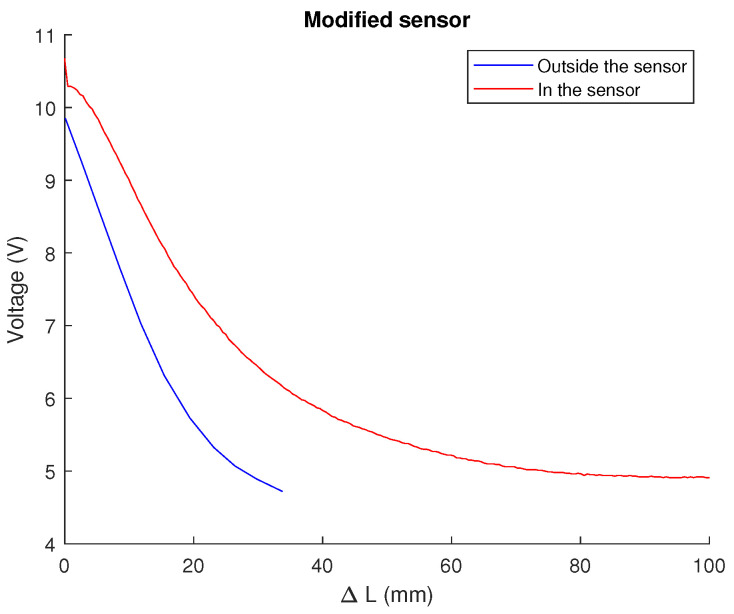
Variation of the behaviour of a sensor placed on the segment (red) versus one whose voltage–length relationship has been measured on the test bench (blue). Source: Authors.

**Figure 8 biomimetics-09-00127-f008:**
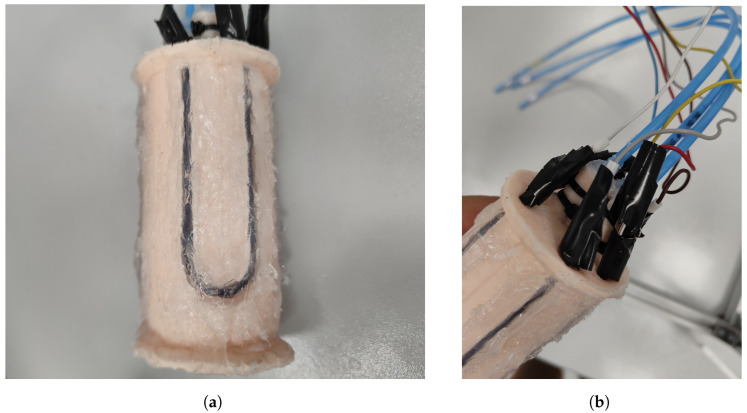
New design of the sensor. (**a**) Segment with bonded sensors using DOWSIL™ 732 sealant. (**b**) Detail of electrical connections and tubing. Source: Authors.

**Figure 9 biomimetics-09-00127-f009:**
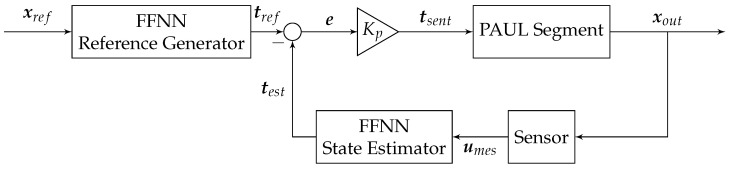
Control loop implemented for a PAUL segment. Source: Authors.

**Figure 10 biomimetics-09-00127-f010:**
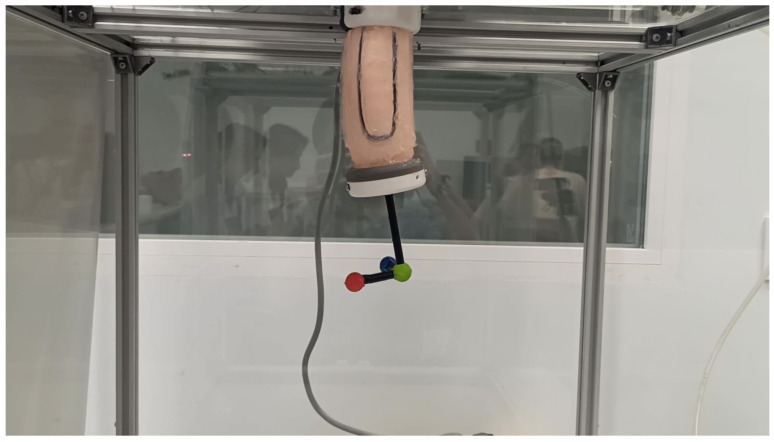
PAUL segment at one step of the data acquisition process. Source: Authors.

**Figure 11 biomimetics-09-00127-f011:**
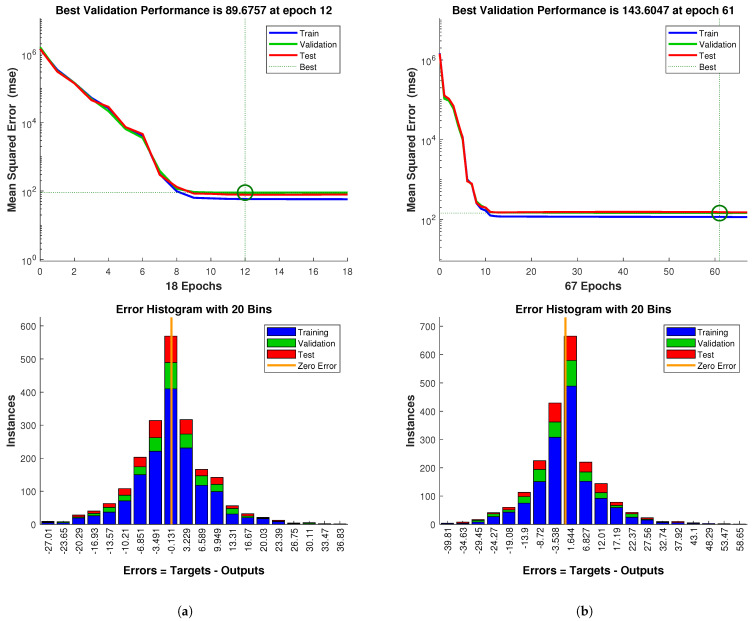
Training curves and error histogram of both nets: (**a**) Reference Generator. (**b**) State Estimator. Errors are expressed in the units of the output variable (ms). The circles in the training curves indicate the period of best performance. From there, the error of the test dataset started to increase. Source: Authors.

**Figure 12 biomimetics-09-00127-f012:**
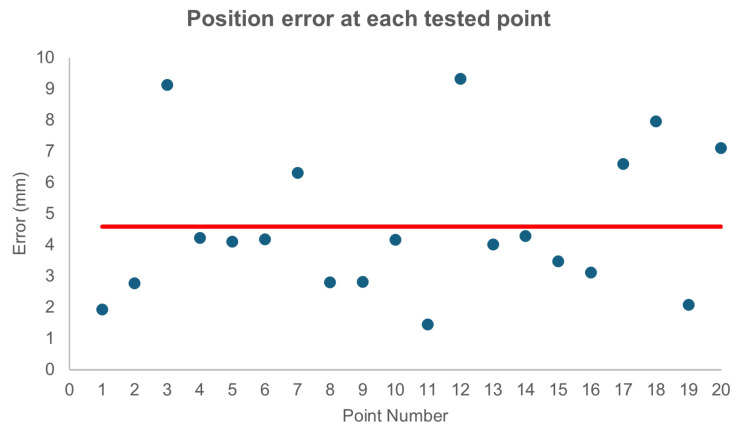
Error (measured in norm 2 and expressed in mm) between the desired position and that achieved by the PAUL segment for the 20 iterations between which no intermediate deflation was performed. It can be seen how the results do not worsen over time, indicating that no errors accumulate. The average error is marked with the red line. Source: Authors.

**Figure 13 biomimetics-09-00127-f013:**
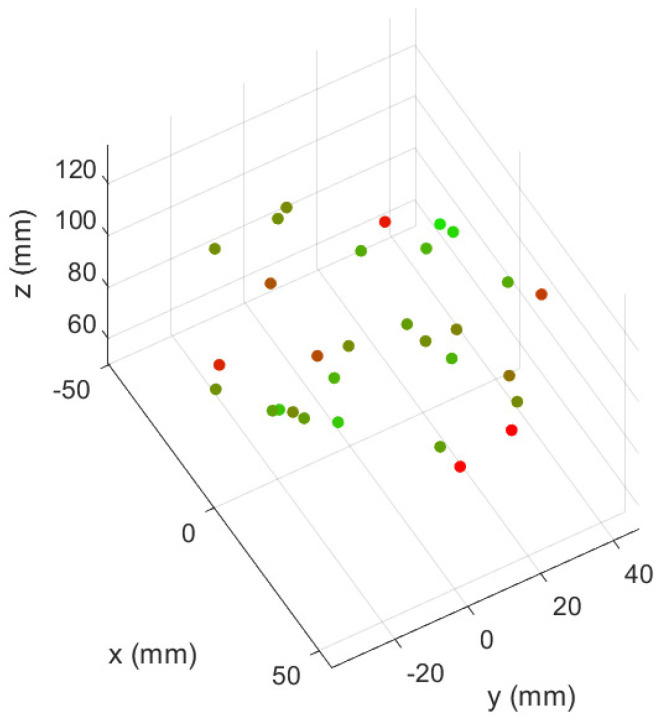
Error (measured in norm 2) between the desired position and that achieved by the PAUL segment for the 20 iterations between which no intermediate deflation was performed. Red dots indicate a larger error than green dots. It can be seen that the control system does not necessarily perform worse in the boundaries of the workspace. Source: Authors.

**Figure 14 biomimetics-09-00127-f014:**
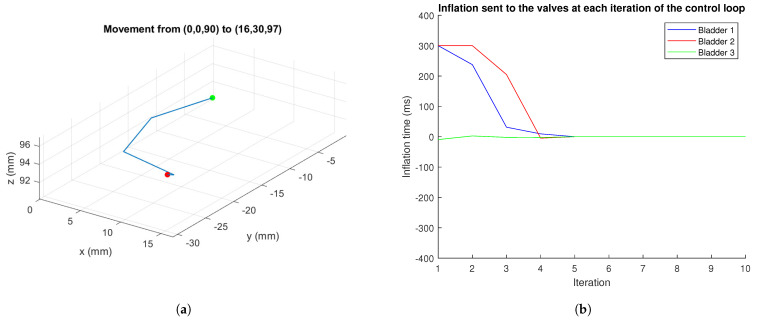
Movement of the robot between two points: (**a**) Intermediate positions reached. (**b**) Control commands sent. Source: Authors.

**Figure 15 biomimetics-09-00127-f015:**
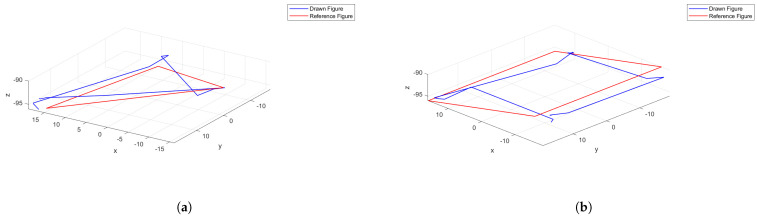
Figure drawing experiments: (**a**) Triangle. (**b**) Square. The red line indicates the figure to draw, while the blue line is the path performed by the PAUL segment. Coordinates are in mm. Source: Authors.

**Figure 16 biomimetics-09-00127-f016:**
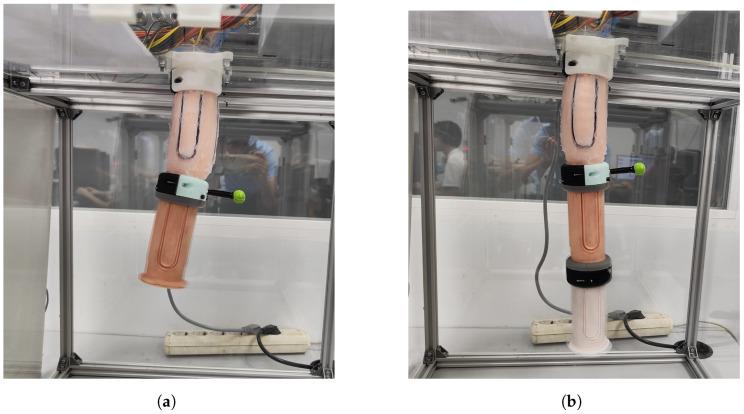
PAUL during the Multiple Segment Experiments. (**a**) With 1 additional segment. (**b**) With 2 additional segments. Source: Authors.

## Data Availability

The data and the code presented in this study are openly available at https://github.com/Robcib-GIT/PAUL (accessed on 8 February 2024).

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
