# Peer review of "Model-Free Control of a Soft Pneumatic Segment"

_biomimetics, 2024, doi:10.3390/biomimetics9030127_

Round 1
Reviewer 1 Report
Comments and Suggestions for Authors
The paper presents a model-free control for a soft actuator (fluidic driven) based on feedback from a stretchable, resistive sensor and an ML model.
The paper is well-written and clearly states all the components, their characterization, and evaluation. Some minor editing issues (e.g., line 483) will help simplify the reading of the paper and thus provide an easier understanding for the audience. A quick proofread of the manuscript will be enough to solve this.
From the technical point of view, there are no significant issues; however, this reviewer has questions about the methodology and the results.
- How many sensors were used? From the pictures, there are three, but there is no evidence of this in the manuscript.
- About the characterization, how do the readings of the sensor change when it is linearly stretched (Figure 3) and when placed in the actuator (Figure 6)? Theoretically, the stretching is not uniform in the second configuration since only a portion of the sensor is stretched. Did the authors use both data sets to validate the ML model or just the ones acquired when integrated into the actuator?
- In Figure 4, the meaning of "modified sensor" is unclear. Is it because of continuous use (how many cycles?) or because of the interaction with the silicone? In the latter case, what do the Authors mean by "modified by the silicone" (line 282)? The reviewer is not asking for a picture at the microscopic level but at least a discussion on why this is happening. Plus, what type of silicone was used? Does this happen with all the silicones? Can the Authors compare the sensor with the silicone and without the silicon after a certain number of cycles?
- The previous question brings us to the following: how repeatable are the measurements at different times? With different sensors?
- Lines 325-327, DOWSIL is commonly used in soft robotic applications for fixing punctures. However, it changes the material's stiffness, making it difficult to replicate the experiments if a different amount of DOWSIL is used. Did the authors compare the actuators' performances before and after using DOWSIL? How replicable are the movements of the actuator if a new one is used?
- Figure 9, what is the meaning of the two circles?
- Section 4.1's results are all related to small displacement; how about large displacement? In this case, the oscillation and the error should be higher. Please consider adding those results to the updated version of the manuscript.
- In Table 1, although the results are remarkable, this reviewer is not convinced that a direct comparison with other methods on different actuators would be the best option; each actuator would act differently according to the shape, number of chambers, and input pressure. For this reason, this reviewer asked about the repeatability of the movement within different batches of actuators. In general, even if using commercial-grade silicone, the manufacturing process is always different, and two actuators might differ in performance (also considering using DOWSIL as a sealant for the resistive sensor). It is probably better to remove it because it is misleading to the readers.
- Figure 11, legend and axis labels need to be included (it is better to add them in the figure, not only in the caption).
- In lines 474-476, it is not possible to directly extend the approach from one segment to multiple without creating a new dataset because the model-free approach cannot consider the structure changes (length, height).
- Lines 479-480, what is the difference in length of the segment when one or more other segments were attached? What is the total weight of the system (before and after)?
Comments on the Quality of English LanguageRephrase line 483
INA is not defined
Author Response
The paper presents a model-free control for a soft actuator (fluidic driven) based on feedback from a stretchable, resistive sensor and an ML model.
The paper is well-written and clearly states all the components, their characterization, and evaluation. Some minor editing issues (e.g., line 483) will help simplify the reading of the paper and thus provide an easier understanding for the audience. A quick proofread of the manuscript will be enough to solve this.
From the technical point of view, there are no significant issues; however, this reviewer has questions about the methodology and the results.
First of all, we would like to thank you for your detailed reading of the article and your valuable comments which have helped us to enrich the paper. We are very pleased that you like it and find it easy to follow.. Your suggestions have been implemented and now appear highlighted in the manuscript. In particular, the English language aspects (e.g. line 483 and the definition of INA) have been corrected.
Response to all of your comments can be find bellow.
- How many sensors were used? From the pictures, there are three, but there is no evidence of this in the manuscript.
Yes, in fact three sensors were used. A sentence explicitly stating this has now been added to the manuscript:
“To implement the control system, the first step was to attach the sensors to the segment. As the segment receives three actuation inputs, three sensors were incorporated.”
- About the characterization, how do the readings of the sensor change when it is linearly stretched (Figure 3) and when placed in the actuator (Figure 6)? Theoretically, the stretching is not uniform in the second configuration since only a portion of the sensor is stretched. Did the authors use both data sets to validate the ML model or just the ones acquired when integrated into the actuator?
Following your comments, Figure 6 has been added showing the relationship between length and voltage in the sensor when it is placed in the actuator. It has been also commented that the existing differences make it difficult to include the data from the linear stretching experiment.
However, even if the behaviour of the sensor were exactly the same, we would find it very difficult to use it on the training dataset because the state observer relates the sensor measurement to inflation times and this latter data cannot be obtained when the sensor is linearly stretched.
- In Figure 4, the meaning of "modified sensor" is unclear. Is it because of continuous use (how many cycles?) or because of the interaction with the silicone? In the latter case, what do the Authors mean by "modified by the silicone" (line 282)? The reviewer is not asking for a picture at the microscopic level but at least a discussion on why this is happening. Plus, what type of silicone was used? Does this happen with all the silicones? Can the Authors compare the sensor with the silicone and without the silicon after a certain number of cycles?
The text has been expanded to include the reviewer's comments. Additionally, it is noted that experimental evidence has shown the variations to be minimal, at least up to 1500 cycles. The authors believe that this small variation does not warrant the inclusion of a figure, but are open to doing so if it would add value to the manuscript.
- The previous question brings us to the following: how repeatable are the measurements at different times? With different sensors?
This discussion has been included.
- Lines 325-327, DOWSIL is commonly used in soft robotic applications for fixing punctures. However, it changes the material's stiffness, making it difficult to replicate the experiments if a different amount of DOWSIL is used. Did the authors compare the actuators' performances before and after using DOWSIL? How replicable are the movements of the actuator if a new one is used?
Experimentally, it has been seen that the points achieved in open loop are similar but not the same. There may be differences of 1 or 2 mm but these are easily addressed with the closed loop control. Similarly, during the work with PAUL, small doses of DOWSIL were added which did not prevent the controller from continuing to function correctly.
As far as the difference between segments is concerned, the very manufacturing of the segments (due to silicone curing, demoulding...) makes them different, never mind even the DOWSIL, which obviously adds another layer of uncertainty.
- Figure 9, what is the meaning of the two circles?
It has been added to the caption.
- Section 4.1's results are all related to small displacement; how about large displacement? In this case, the oscillation and the error should be higher. Please consider adding those results to the updated version of the manuscript.
The errors for each assessed point in section 4.1 have been plotted. The plot shows high and low error points at both the border and centre of the workspace, suggesting that the model is not only valid for small displacements.
In these tests, swelling was limited to 1000ms, which is considered adequate to prevent leakage in normal operation mode, despite occasional slight increases. As stated in the paragraph where the figure is presented, the resulting working space measures approximately 100mm on each side.
- In Table 1, although the results are remarkable, this reviewer is not convinced that a direct comparison with other methods on different actuators would be the best option; each actuator would act differently according to the shape, number of chambers, and input pressure. For this reason, this reviewer asked about the repeatability of the movement within different batches of actuators. In general, even if using commercial-grade silicone, the manufacturing process is always different, and two actuators might differ in performance (also considering using DOWSIL as a sealant for the resistive sensor). It is probably better to remove it because it is misleading to the readers.
Thank you for your thoughts. From the beginning of the manuscript writing process, we were very hesitant about whether or not to include it because, as you pointed out, each actuator performs very differently. Following your suggestions, we have removed it, leaving only a textual comparison with other actuators and control methods.
- Figure 11, legend and axis labels need to be included (it is better to add them in the figure, not only in the caption).
It has been corrected.
- In lines 474-476, it is not possible to directly extend the approach from one segment to multiple without creating a new dataset because the model-free approach cannot consider the structure changes (length, height).
Yes, a new dataset is needed. However, as mentioned in the last paragraph before the conclusions, we think that Transfer Learning or the implementation of two neural networks in series may allow to use a second dataset much smaller in size. If 150-200 points are enough, the robot should be ready in 15 minutes.
- Lines 479-480, what is the difference in length of the segment when one or more other segments were attached? What is the total weight of the system (before and after)?
This data has been included in the manuscript:
With one segment, PAUL is 125mm long and weighs 187g. When one segment is added, it becomes 250mm and 375g, and 375mm and 560g when two segments are added.
Reviewer 2 Report
Comments and Suggestions for Authors
Modelling of soft robotics has long been challenging in this field. This work tried to model the soft segment using neural network or data-driven method. The work is quite interesting and will offer readers valuable information. However, I have some suggestions to help the authors to improve the paper:
1. It is claimed that the PAUL Approach. But there is no explanation of it.
2. the quality of the developed prototype of soft segment should be improved.
Comments on the Quality of English Languagequite good.
Author Response
Modelling of soft robotics has long been challenging in this field. This work tried to model the soft segment using neural network or data-driven method. The work is quite interesting and will offer readers valuable information. However, I have some suggestions to help the authors to improve the paper:
We sincerely appreciate your time and effort in reviewing our paper, and we are grateful for your positive feedback on the content.
- It is claimed that the PAUL Approach. But there is no explanation of it.
Although reference [15] (of the original manuscript) presents an article introducing PAUL (in the original manuscript an arXiv preprint was cited, in this version an already published paper), a more detailed explanation of the robot and its operation has been introduced, following your suggestion.
- the quality of the developed prototype of soft segment should be improved.
Indeed, we believe that it is necessary, in the long term, to improve the quality of the prototype. This is a first design on which we are working to improve. In particular, we are looking for ways to build more repeatable and robust modules. One option we are considering is to manufacture it with silicon 3D printing. This possible future line is now discussed in the section describing PAUL.
Reviewer 3 Report
Comments and Suggestions for Authors
The paper tackles a nice topic: a simple, low-cost biologically inspired soft robot.
The authors are aware of the huge research field and already existing robots/analyses/prototypes and present their results ... always compared to the literature. They sparre no effort on a huge literature overview in several sections. A real good job.
Thank you for the ability of reading this paper.
I am really surprised about the modular structure of your robot.
At the end, the authors demonstrate the effectiveness of their robot in a few examples. Point to point movement and "figure drawing". What about reproducibility?
A short hint:
- Figure 2: What are we able to see there? Sorry, but the figure looks a bit funny. A black, thick, solid line.
Author Response
The paper tackles a nice topic: a simple, low-cost biologically inspired soft robot.
The authors are aware of the huge research field and already existing robots/analyses/prototypes and present their results ... always compared to the literature. They sparre no effort on a huge literature overview in several sections. A real good job.
Thank you for the ability of reading this paper.
I am really surprised about the modular structure of your robot.
We very much appreciate the comments received. We are pleased that the work presented here is to your liking.
At the end, the authors demonstrate the effectiveness of their robot in a few examples. Point to point movement and "figure drawing". What about reproducibility?
In each of the three point-to-point experiments, the robot has been sent to a total of 20 different positions. Although this is not a very large number, it was considered sufficient, since, as shown in Figure 10, no drift or increase in error was observed as the robot was moved to different points.
As for figure drawing, several more tests were carried out. In all of them the errors and shapes were similar, so they were not included as it was believed that they did not provide relevant information for the article. In view of the comments, the new error results have been commented on.
A short hint:
- Figure 2: What are we able to see there? Sorry, but the figure looks a bit funny. A black, thick, solid line.
It has been exchanged for a more appropriate sensor image.
Reviewer 4 Report
Comments and Suggestions for Authors
The paper presents a model-free control of a soft pneumatic segment. The proposed approach seems interesting. However, the following issues should be considered to improve the paper's contributions.
1. The research appears to be more akin to a review paper discussing sensors and control strategies rather than original research papers.
Subsections 3.1 and 3.2: An essential aspect of sensor usage pertains to verifying their accuracy. On page 8, line 282, the authors claim that the modified sensor exhibits improved performance. However, it is crucial to elucidate how the superiority of the modified sensor is verified. It is my belief that the sensor serves as the most accurate instrument in the system. Therefore, employing more precise devices to validate its performance is imperative.
3. The control requirement must be provided to verify whether the proposed controller is met it or not.
4. Resembles comment number 2. On page 6, line 260, the authors mention the utilization of a 2 mm accurate sensor. However, in lines 414 to 417, the authors state, "An error of 4.27 mm with a standard deviation of 2.67 mm was obtained using norm 2. For data involving intermediate deflation, errors reduced to 3.94 mm with a standard deviation of 1.25 mm. Conversely, for the second set of test data, the error was 4.59 mm with a standard deviation of 2.35 mm." How can the authors demonstrate a precision of "0.01 mm" when the sensor's accuracy is only 2 mm? Kindly elucidate, or clarify if the reviewer has misunderstood.
5. The control output of the proposed controller should be provided.
6. Additional experimental scenarios should be explored. For instance, integrating load or external disturbances into the system is necessary.
Author Response
The paper presents a model-free control of a soft pneumatic segment. The proposed approach seems interesting. However, the following issues should be considered to improve the paper's contributions.
First of all, we would like to thank you for your detailed reading of the article and your valuable comments which have helped us to enrich the paper. Your suggestions have been implemented and now appear highlighted in the manuscript.
- The research appears to be more akin to a review paper discussing sensors and control strategies rather than original research papers.
We are sorry that this article may have seemed like this at first glance. The aim has been to provide a very detailed analysis of the existing literature so far in order to show that the contribution is correctly framed and to demonstrate that it is truly valuable. The article does, however, provide two interesting novelties, namely the sensorisation of the robot and the implementation of a high-precision closed-loop control system.
To emphasise this and so that the article does not look like a survey, the contributions made have been reformulated in the introduction and Table 1 has been removed from the Results 1 section, commenting on the values obtained without going into so much comparison with the work of other authors.
Subsections 3.1 and 3.2: An essential aspect of sensor usage pertains to verifying their accuracy. On page 8, line 282, the authors claim that the modified sensor exhibits improved performance. However, it is crucial to elucidate how the superiority of the modified sensor is verified. It is my belief that the sensor serves as the most accurate instrument in the system. Therefore, employing more precise devices to validate its performance is imperative.
The modified sensor is better because, on a qualitative level, it offers better properties, mainly lower hysteresis, and higher measuring range, as can be seen in Figure 4. To emphasise that this is an improvement in properties, it has been commented in Figure 4 caption.
To measure length and voltage in the sensor, the instrumentation shown in Figure 3 was used, consisting of a rack and pinion mechanism connected to a stepper motor that was lengthening the sensor and an INA 3221, which measured voltages with a resolution of 0.01V. As the stretching lengths were controlled at all times, it was possible to obtain a faithful portrait of the sensor's behaviour.
- The control requirement must be provided to verify whether the proposed controller is met it or not.
It has been included at the end of Section 2:
The main objective of this work is to achieve closed-loop kinematic control of one of its segments with a point-to-point error less than 5 mm. If this requirement is met, the following objective will be to evaluate this control objective when the weight of additional segments is carried by the robot.
Additionally, in the Conclusion, a discussion about if the control requirement has been fulfilled has been addressed:
The objective of the control can be considered fulfilled. Furthermore, the accuracy of the proposed controller is demonstrated by comparing the results with those achieved by other soft robots.
- Resembles comment number 2. On page 6, line 260, the authors mention the utilization of a 2 mm accurate sensor. However, in lines 414 to 417, the authors state, "An error of 4.27 mm with a standard deviation of 2.67 mm was obtained using norm 2. For data involving intermediate deflation, errors reduced to 3.94 mm with a standard deviation of 1.25 mm. Conversely, for the second set of test data, the error was 4.59 mm with a standard deviation of 2.35 mm." How can the authors demonstrate a precision of "0.01 mm" when the sensor's accuracy is only 2 mm? Kindly elucidate, or clarify if the reviewer has misunderstood.
This line has been reworded to avoid confusion. The 2 mm indicated does not refer to the accuracy of the sensor, but to its diameter. We are sorry for the inconvenience caused and we sincerely thank you for your comment, which has helped to clarify this crucial point and prevent future misunderstandings by the readers if the manuscript is finally published.
- The control output of the proposed controller should be provided.
It has been included in Figure 13.
- Additional experimental scenarios should be explored. For instance, integrating load or external disturbances into the system is necessary.
Although there is no test where weights are specifically tested as is done in other works, and even in PAUL's own previous work, it has not been considered necessary to do so since the addition of additional segments is still a test of the behaviour under weight. To emphasise this, the increase in weight of the addition of each new segment has been added.
Round 2
Reviewer 4 Report
Comments and Suggestions for Authors After reviewing the revised manuscript and the authors' responses, I find that they have effectively addressed the raised concerns. I agree with the explanations provided by the authors and believe that the manuscript is now suitable for publication in Biomimetics.